# From Pandemic to Resistance: Addressing Multidrug-Resistant Urinary Tract Infections in the Balkans

**DOI:** 10.3390/antibiotics14090849

**Published:** 2025-08-22

**Authors:** Rumen Filev, Boris Bogov, Mila Lyubomirova, Lionel Rostaing

**Affiliations:** 1Department of Nephrology, Internal disease Clinic, University Hospital “Saint Anna”, 1709 Sofia, Bulgaria; bbogov@yahoo.com (B.B.); mljubomirova@yahoo.com (M.L.); 2Faculty of Medicine, Medical University Sofia, 1504 Sofia, Bulgaria; 3Nephrology, Hemodialysis, Apheresis and Kidney Transplantation Department, Grenoble University Hospital, 38700 Grenoble, France; 4Internal Disease Department, Grenoble Alpes University, 38400 Grenoble, France

**Keywords:** multidrug-resistant urinary tract infections (MDR UTIs), antimicrobial resistance (AMR), Balkans, COVID-19 pandemic, health policy

## Abstract

**Background/Objectives**: The rise in urinary tract infections caused by multidrug-resistant (MDR) bacteria presents a serious public health challenge across the Balkans, a region already burdened by aging populations, healthcare resource limitations, and fragmented antimicrobial surveillance systems. **Methods**: This review explores the epidemiology, risk factors, and consequences of MDR UTIs, particularly in the context of the COVID-19 pandemic, which significantly accelerated antimicrobial resistance (AMR) due to widespread, inappropriate antibiotic use. **Results**: The paper discusses region-specific data on resistance trends, highlights the gaps in diagnostic infrastructure, and evaluates emerging clinical strategies including antimicrobial stewardship (AMS), rapid diagnostic technologies, novel antibiotics, and non-antibiotic alternatives such as bacteriophage therapy and vaccines. **Conclusions**: Policy recommendations are provided to strengthen surveillance, promote evidence-based treatment, and ensure equitable access to diagnostic and therapeutic tools. A multidimensional and regionally coordinated response is essential to curb the MDR UTI burden and safeguard public health across the Balkans.

## 1. Introduction

Multidrug-resistant (MDR) urinary tract infections (UTIs) represent an escalating public health challenge in the Balkans, placing considerable strain on healthcare systems due to limited therapeutic options and increased morbidity. This issue has become particularly pronounced following the onset of the COVID-19 pandemic, which has further exacerbated the burden on healthcare infrastructure. The frequent and often inappropriate use of antibiotics remains a key contributing factor. Concurrently, the aging population in the region exerts additional pressure on healthcare resources and negatively influences treatment outcomes. These combined factors adversely affect the success rates of various therapeutic interventions, including those targeting UTIs.

### Epidemiology of MDR UTIs

MDR UTIs are an increasingly prevalent concern in the Balkans. Globally, UTIs rank among the most common bacterial infections, affecting individuals across all age groups. Women are disproportionately affected, with nearly half experiencing at least one UTI in their lifetime [1]. The burden is also considerable among children, the elderly, and individuals with underlying conditions such as diabetes or chronic immunosuppression [2].

UTIs are typically categorized as either uncomplicated or complicated. Complicated UTIs are associated with factors such as anatomical or functional abnormalities of the urinary tract, catheterization, or chronic comorbidities [3]. Escherichia coli is the predominant pathogen, responsible for approximately 70–95% of community-acquired infections. Other common uropathogens include *Klebsiella* spp., *Proteus* spp., and *Enterococcus* spp. [3].

The increasing prevalence of antibiotic-resistant uropathogens poses a serious global health threat. The emergence of extended-spectrum beta-lactamase (ESBL)-producing bacteria and other multidrug-resistant organisms (MDROs) has complicated treatment approaches, leading to increased healthcare costs and patient morbidity [2]. A comprehensive understanding of UTI epidemiology is essential for developing effective prevention strategies, guiding empirical therapy, and curbing the spread of resistant pathogens [1].

## 2. Results

### 2.1. Risk Factors for MDR UTIs

Numerous developments over the past decade have contributed to the growing prevalence of multidrug-resistant (MDR) urinary tract infections (UTIs). While some of these risk factors initially appeared to be independent, they have become increasingly interrelated over time. One of the most prominent contributors is the frequent and inappropriate use of antibiotics. Studies indicate that individuals who have used antibiotics three or more times are at significantly increased risk of developing MDR UTIs compared to those with no recent antibiotic use [4]. Notably, the use of fluoroquinolones and tetracyclines has been strongly associated with infections caused by extended-spectrum beta-lactamase (ESBL)-producing Escherichia coli [5]. Additional independent risk factors include male sex, advanced age, and a history of UTI within the previous six months [4].

The overuse of antibiotics during the COVID-19 pandemic has played a substantial role in accelerating the emergence of MDR UTIs, a trend that appears particularly marked in the Balkan region. Several studies have demonstrated a direct correlation between increased antibiotic prescribing during the pandemic and the subsequent rise in antimicrobial resistance (AMR) [6,7,8]. A systematic review encompassing 892,312 patients diagnosed with COVID-19 across 173 studies revealed that 76.2% received antibiotic therapy, despite the relatively low prevalence of bacterial co-infections [6]. This indiscriminate use of antibiotics, especially in low- and middle-income countries, has facilitated the selection and spread of MDR organisms, including ESBL-producing bacteria, which are commonly implicated in UTIs.

In a retrospective study conducted at the University General Hospital of Alexandroupolis in Greece, a notable increase in antimicrobial resistance was observed during the pandemic period [7]. The study documented a rise in infections caused by carbapenem-resistant Enterobacterales, a group frequently implicated in difficult-to-treat UTIs. Similarly, a multicenter study from Serbia reported that 12.9% of hospitalized COVID-19 patients developed secondary bacterial infections, most of which were hospital-acquired [8]. Among the predominant pathogens were Klebsiella pneumoniae and Acinetobacter baumannii, both of which demonstrated high resistance to carbapenems. The study further noted that patients over the age of 60 and those with polymicrobial infections exhibited higher mortality rates [8].

According to the European Centre for Disease Prevention and Control’s Surveillance Atlas of Infectious Diseases, antimicrobial resistance has significantly increased in Bulgaria since 2019 [9]. For example, combined resistance of Klebsiella pneumoniae to third-generation cephalosporins, fluoroquinolones, and aminoglycosides rose from 44.9% in 2019 to 64.9% in recent years (Figure 1) [9]. This upward trend is similarly observed across other major uropathogens. A study by Mares et al. also reported a worsening resistance profile in Romania during the COVID-19 pandemic, highlighting increased resistance across multiple antibiotic classes commonly used for UTIs [10].

The implications of increasing antimicrobial resistance are multifaceted. These include higher healthcare costs, prolonged hospital stays, and elevated mortality rates [11]. Another retrospective study comparing resistance patterns of uropathogens before and during the pandemic revealed significant increases among Gram-negative bacteria. Notably, resistance in *Klebsiella* spp. to quinolones increased from 16.87% to 35.51%, and in *Pseudomonas* spp. from 30.3% to 77.41% [12].

The widespread administration of antibiotics to patients with suspected or confirmed SARS-CoV-2 infection has further aggravated the AMR crisis. Although antibiotics do not treat viral infections, they were often prescribed during the early phases of the pandemic due to diagnostic uncertainty and concern about bacterial co-infections. Early assessments [13,14,15,16] showed that a large proportion of antibiotic prescriptions were issued to patients with COVID-19, even though few of these patients had confirmed bacterial infections—particularly outside intensive care units. This trend was not limited to the Balkans but was observed globally.

A recent survey conducted in public and private clinics in Ireland found that 76% of participants reported no beneficial effect from antibiotic therapy in COVID-19 patients [17]. In another study by Abdel Gawad et al., a significant increase in resistance was observed in the majority of tested bacterial strains during the pandemic [18]. Research from Morocco published in February 2023 assessed resistance patterns in uropathogens before and after the COVID-19 outbreak and found marked increases in resistance—especially in E. coli (to amoxicillin and levofloxacin), *Klebsiella* spp. (to amoxicillin and ceftriaxone), and *Enterococcus* spp. (to levofloxacin and ciprofloxacin). Interestingly, the same study noted decreased resistance to amikacin, carbapenems, and trimethoprim-sulfamethoxazole in *Klebsiella* spp., and to ceftriaxone, carbapenems, and trimethoprim-sulfamethoxazole in *Enterococcus* spp.—findings consistent with those of our own analysis [19]. A comparative study from Iran further highlighted the pandemic’s impact, demonstrating significant increases in resistance among E. coli to ampicillin, carbapenems, and ceftazidime, and among *Klebsiella* spp. to ampicillin, levofloxacin, and ceftazidime between 2020 and 2022 [20].

### 2.2. Surveillance and Control Strategies

Despite growing awareness of antimicrobial resistance (AMR) and the need for a coordinated public health response, surveillance systems across the Balkan region remain highly fragmented and inconsistent. Some countries, such as Greece and Romania, have made significant progress through their active participation in the European Antimicrobial Resistance Surveillance Network (EARS-Net) and the WHO/Europe Central Asian and Eastern European Surveillance of Antimicrobial Resistance (CAESAR) initiative [21,22]. These programs enable standardized data collection and facilitate cross-national comparisons. Greece and Romania, for example, routinely report AMR data from blood and cerebrospinal fluid isolates, providing a relatively robust understanding of resistance trends—particularly for critical pathogens such as Escherichia coli and Klebsiella pneumoniae.

However, surveillance coverage remains uneven throughout the region. Several Western Balkan countries, including Albania, Bosnia and Herzegovina, North Macedonia, and Montenegro, face persistent challenges. These include limited microbiological capacity, a lack of automated diagnostic systems, shortages of trained personnel, and poor integration between hospital laboratories and national public health institutions [23]. In some cases, surveillance is based solely on data from a few sentinel sites or tertiary hospitals, which may not reflect the broader population or regional variability [23]. Moreover, smaller countries may struggle to secure adequate funding or technical infrastructure to implement international guidelines consistently or to submit timely and comprehensive data reports.

The lack of harmonized surveillance data significantly hampers effective regional benchmarking and coordinated response strategies. For instance, while Greece has submitted detailed AMR data to EARS-Net over the past decade, neighboring countries such as Bulgaria, North Macedonia, Albania, and Bosnia and Herzegovina have provided only limited or irregular datasets [24]. This inconsistency limits the ability to conduct meaningful cross-national comparisons. Additionally, delays in data reporting and the use of outdated antimicrobial susceptibility breakpoints in some laboratories undermine the reliability of resistance trend analyses and hinder timely public health responses.

The COVID-19 pandemic has further exposed and exacerbated these vulnerabilities. The widespread use of antibiotics—often without confirmed bacterial infections—has likely intensified resistance pressures across the region [25]. Simultaneously, the pandemic disrupted routine laboratory workflows, reducing both the quantity and quality of samples available for surveillance [26].

Thus, while certain strides have been made in AMR surveillance within the Balkans, the region continues to face substantial obstacles, including fragmented reporting systems, data inaccessibility, and critical infrastructural limitations. These challenges severely limit the capacity to effectively monitor and control the growing threat of multidrug-resistant infections—particularly urinary tract infections caused by Gram-negative pathogens such as *E. coli* and Klebsiella pneumoniae.

Future efforts must focus on strengthening laboratory networks, investing in workforce training and modern diagnostic equipment, and fostering regional collaboration through unified surveillance platforms. Such initiatives will be essential to improving the detection, treatment, and prevention of MDR UTIs across the Balkans.

### 2.3. Clinical and Public Health Implications of Multidrug-Resistant Urinary Tract Infections in the Balkans

Multidrug-resistant (MDR) urinary tract infections (UTIs) represent a growing clinical and public health burden in the Balkan region [27]. The emergence of resistant uropathogens—particularly ESBL-producing Escherichia coli, carbapenem-resistant Klebsiella pneumoniae, and Acinetobacter baumannii—has significantly compromised the efficacy of conventional empirical treatment regimens [27,28]. As a result, clinicians increasingly rely on last-resort antibiotics such as colistin, meropenem, and imipenem. However, these agents are associated with increased toxicity, higher costs, and limited effectiveness in elderly or immunocompromised patients [29,30].

The clinical consequences of MDR UTIs include prolonged hospitalization, delays in the initiation of appropriate therapy, and elevated risks of complications such as pyelonephritis, urosepsis, and infection recurrence. A recent multicenter study from Serbia reported that over 12% of hospitalized patients with confirmed COVID-19 developed secondary bacterial infections, with high resistance rates contributing to increased mortality—particularly among patients over 60 years of age and those with multiple comorbidities [31]. Similarly, retrospective analyses from Romania and Bulgaria have shown that MDR UTIs often lead to longer inpatient stays and higher healthcare expenditures, largely due to the need for intravenous therapy, isolation precautions, and repeated diagnostic evaluations [21,29].

The public health implications of MDR UTIs extend well beyond individual clinical settings. The widespread reliance on second- and third-line therapies places a significant financial strain on already under-resourced healthcare systems. In many Balkan countries, the combination of an aging population, underfunded health infrastructure, and limited antimicrobial stewardship has created a cycle of overtreatment, resistance propagation, and therapeutic failure [26,32]. The disruption of routine microbiological surveillance during the COVID-19 pandemic further worsened this dynamic by reducing both the quantity and quality of bacterial isolates submitted for antimicrobial resistance (AMR) monitoring [26].

In response to these challenges, a therapeutic study was conducted at our institution targeting patients with recurrent MDR UTIs who had failed two or more prior antibiotic regimens. The causative pathogen in this cohort was Klebsiella pneumoniae exhibiting multidrug resistance. While initial treatment was guided by antibiogram results, a subsequent tailored regimen based on sulfamethoxazole/trimethoprim administered over an 11-week protocol proved to be effective, cost-efficient, and well-tolerated [33]. This case exemplifies the potential of targeted, resource-conscious antibiotic strategies in the post-pandemic era.

In this context, antimicrobial stewardship (AMS) has emerged as one of the most effective and sustainable approaches to address the growing threat of MDR UTIs—particularly in resource-constrained healthcare systems such as those in the Balkans [34,35]. A comprehensive post-pandemic review published on ScienceDirect demonstrated that even short-term AMS interventions, when well-structured and implemented, significantly reduced the incidence of MDR organisms, especially in patients with complicated or recurrent UTIs [34]. These programs not only preserved clinical outcomes but also decreased antibiotic consumption and healthcare spending. Core elements of successful AMS initiatives included establishing baseline antibiotic use metrics, conducting regular surveillance of local resistance patterns, providing targeted prescriber education, and integrating electronic alert systems to guide evidence-based prescribing [34].

Further evidence supports the efficacy of AMS programs that combine real-time prescribing feedback with institutional policies to reduce the empirical use of broad-spectrum antibiotics—a practice that became widespread during the diagnostic uncertainty of the pandemic [36]. Such strategies improve patient outcomes while generating substantial cost savings by minimizing unnecessary hospitalizations, the use of expensive second-line agents, and the complications associated with resistant infections. During the COVID-19 crisis, AMS programs played a crucial role in mitigating inappropriate antibiotic use and preserving the effectiveness of available treatments [37].

Global health authorities, including the World Health Organization and the European Centre for Disease Prevention and Control, now recognize AMS as a cornerstone strategy in the fight against AMR. They have advocated for its widespread adoption, particularly in low- and middle-income countries, through the development of tailored toolkits, national policy frameworks, training programs, and digital infrastructure support [38].

In summary, AMS programs, when combined with diagnostic stewardship and institutional commitment—offer a scalable, evidence-based approach to managing the increasing burden of MDR UTIs in the post-pandemic landscape. Their effectiveness in improving prescribing practices, containing costs, and limiting resistance makes them indispensable tools for healthcare systems across the Balkans and beyond. Drawing upon the lessons of recent years, we have formulated an alternative therapeutic strategy grounded in current evidence and contextual realities—providing a framework adapted to the distinct challenges of the post-COVID era [33].

### 2.4. The Benefits of Rapid Diagnostic Testing and Therapy—New Approaches

Recent advances in point-of-care diagnostic technologies, most notably the Sysmex Astrego PA-100 AST System—have revolutionized UTI management by drastically reducing diagnostic and antibiotic susceptibility testing (AST) turnaround times. This nanofluidic, optical-based platform can detect bacteriuria in just 15 min and deliver a full antibiotic susceptibility profile within 30–45 min, in stark contrast to the 24–72 h required for conventional urine cultures and laboratory-based AST [39]. Clinical evaluations have demonstrated strong performance: a near-patient study involving 278 women reported 84% sensitivity and 99% specificity for bacteriuria detection, with accurate pathogen–antibiotic matching in approximately 78% of cases. Notably, the rate of optimal antimicrobial selection increased from 58% (standard care) to 78% using the PA-100 system [40]. A 2025 budget impact analysis in Spain projected first-year savings of over €323 million (≈€119 per patient), largely driven by reductions in inappropriate antibiotic use, complications, and productivity loss [41]. By enabling rapid, targeted narrow-spectrum therapy at the bedside, PA 100 helps reduce empirical broad-spectrum prescribing and supports core antimicrobial stewardship (AMS) goals—limiting drug exposure, minimizing resistance development, and generating both clinical and economic benefits [40,41].

From a public health perspective, these diagnostics are critical tools within AMS programs. They bridge the gap between diagnosis and stewardship, reduce diagnostic uncertainty, and promote evidence-based treatment decisions—even in outpatient or primary-care settings. Their use aligns with WHO and ECDC recommendations emphasizing the importance of rapid diagnostics in combating AMR. In the Balkans, where empirical prescribing remains widespread and surveillance systems are often fragmented, adopting such technologies could dramatically reduce inappropriate antibiotic use, improve patient outcomes, and ease the economic burden of AMR. Their implementation should be considered a high priority.

Recent developments in antibiotic therapy also offer new opportunities to address MDR UTIs, particularly when integrated with diagnostic and stewardship strategies. Cefiderocol, a first-in-class siderophore cephalosporin, demonstrated non-inferiority to imipenem–cilastatin in the APEKS-cUTI Phase III trial for complicated UTIs. It showed high rates of microbiological eradication and clinical cure in hospitalized patients infected with Gram-negative MDR pathogens [42,43]. Its unique “Trojan horse” mechanism facilitates entry into bacterial cells by exploiting iron transport channels, bypassing many resistance mechanisms.

Gepotidacin (Blujepa), an oral triazaacenaphthylene antibiotic with a novel dual-target mechanism (inhibiting both DNA gyrase and topoisomerase), was shown in the EAGLE-2 and EAGLE-3 trials to be non-inferior—and in one trial, superior—to nitrofurantoin for uncomplicated UTIs in adults and adolescents. Both trials demonstrated favorable composite microbiological and clinical cure rates [44].

Pivmecillinam (Pivya), a narrow-spectrum penicillin derivative long used in Europe, was recently approved by the FDA. Clinical trials comparing it to placebo, cephalexin, and ibuprofen reported composite cure rates ranging from 62.0% to 71.7%, with clinical cure rates between 63.5% and 82.7%, and microbiological success ranging from 74.3% to 86.9% at test-of-cure (Days 7–15) [45]. Pivmecillinam’s targeted activity—primarily against E. coli, Proteus mirabilis, and Staphylococcus saprophyticus—makes it an ideal, cost-effective first-line agent that aligns with AMS priorities by reducing the need for broad-spectrum antibiotics.

Non-antibiotic alternatives are also gaining traction in the context of rising AMR. Bacteriophage therapy has shown promising results, with a 2023 systematic review reporting that over 72% of included studies demonstrated microbiological and clinical improvement in UTI patients treated with phages [46]. A 2024 study further demonstrated phages’ ability to penetrate biofilms and enhance antibiotic efficacy, making them particularly useful in complex or refractory infections [47]. Although clinical trials remain limited, early-phase studies have reported good safety profiles and significant infection resolution [48].

UTI vaccines have also shown substantial potential in reducing recurrence. The oral immunostimulant OM-89 (UroVaxom), a lyophilized E. coli extract, was evaluated in a multicenter, double-blind trial that found a 34% reduction in UTI incidence over 12 months compared to placebo (0.84 vs. 1.28 episodes per patient-year; *p* < 0.003) [49]. A systematic review in European Urology Focus supported these findings and confirmed OM-89’s inclusion in current European guidelines for recurrent UTI prevention [50].

Interest has also grown around StroVac (Solco-Urovac), a polyvalent, inactivated bacterial vaccine administered intramuscularly in a three-dose regimen with annual boosters. In a large European double-blind, placebo-controlled trial involving 376 adults with recurrent UTIs, StroVac reduced the average number of infections from 5.5 to 1.2 over a 13.5-month follow-up. Although the primary efficacy endpoint was not statistically significant (*p* = 0.63), a subgroup with ≥7 UTIs annually showed a significant reduction (from 7.3 to 2.3 vs. 7.6 to 4.4; *p* = 0.048) [51]. A comparative observational study found that 79.3% of StroVac recipients remained UTI-free after two years, compared to 59.2% in those receiving nitrofurantoin prophylaxis (*p* = 0.03), with few adverse events [52]. Preclinical data in murine models showed StroVac activated innate immunity—including macrophage phagocytosis and cytokine production—indicating that both innate and adaptive immune mechanisms may underlie its effects [53]. Although definitive superiority over placebo remains to be confirmed, StroVac represents a promising prophylactic strategy, particularly for patients at high risk of recurrent MDR UTIs.

Together, these emerging tools—rapid diagnostics, novel antibiotics, bacteriophage therapy, and vaccines—represent a paradigm shift in the prevention and management of MDR UTIs. They target critical stages in the infection pathway, such as pathogen detection, antibiotic selection, bacterial adhesion, biofilm formation, and immune modulation. Their integration offers a strategic opportunity to reduce reliance on broad-spectrum antibiotics and enhance the effectiveness of AMS initiatives.

However, despite growing global evidence, these innovations remain largely underutilized in many parts of the Balkans. In Bulgaria, for instance, empirical antibiotic prescribing continues to dominate clinical practice, even in cases of recurrent or complicated UTIs—often without microbiological confirmation. Non-antibiotic strategies such as phage therapy or UTI vaccination are rarely discussed in primary-care settings, limited by a lack of awareness, regulatory challenges, and restricted access. Additionally, a cultural expectation that antibiotics represent the standard of care further reinforces prescribing inertia.

This persistent overreliance on antibiotics continues to fuel the region’s AMR crisis and undermines long-term infection control efforts. Broader integration of non-antibiotic and precision-based strategies—supported by clinician education, health policy reform, and investment in infrastructure—could provide a more sustainable, cost-effective solution for UTI management in the Balkans and beyond.

### 2.5. Policy Recommendations and Implementation of Roadmap for the Balkans

Effectively addressing the growing burden of multidrug-resistant urinary tract infections (MDR UTIs) in the Balkans requires more than clinical advancements; it necessitates coordinated, evidence-based policy action. Although modest progress has been made in areas such as surveillance and antimicrobial stewardship (AMS), systemic gaps remain. To address these shortcomings, we propose a structured roadmap for national and cross-border implementation of effective UTI control strategies in the region.

We would like to address the following points:1.Establish a Regional AMR Taskforce:

A dedicated taskforce should be formed under the auspices of the WHO/Europe CAESAR network or the European Centre for Disease Prevention and Control (ECDC). This body would coordinate national strategies, harmonize surveillance protocols, and facilitate cross-border data sharing. Countries such as Greece and Romania—already active contributors to EARS-Net—could serve as regional models [21,22,24]. Joint training initiatives, capacity building, and the standardization of diagnostic methodologies would significantly enhance interoperability and collective preparedness.

2.Mandatory surveillance and real-time data integration

Surveillance systems must expand beyond tertiary-care centers to encompass primary-care clinics, nursing homes, and private laboratories. National mandates requiring data submission to EARS-Net or CAESAR should be enforced and supported by investment in electronic health record (EHR) integration and automated reporting platforms [23,24]. Real-time antibiogram updates should inform empirical prescribing practices, thereby reducing unnecessary use of broad-spectrum antibiotics.

3.Investment in rapid diagnostic technologies

Governments should provide financial incentives and subsidies to facilitate the deployment of point-of-care diagnostic tools such as the Sysmex PA-100, particularly in outpatient and underserved rural settings. These technologies dramatically reduce diagnostic turnaround times, enable timely and targeted therapy, and align with AMS principles by minimizing empirical broad-spectrum antibiotic use [39,40,41].

4.Strengthen AMS programs in hospitals and outpatient clinics

Healthcare institutions should be mandated to implement AMS teams composed of infectious disease physicians, clinicians, clinical microbiologists, and pharmacists. These teams would be responsible for monitoring prescribing trends, conducting audit-and-feedback cycles, and promoting de-escalation based on culture and susceptibility data. Even short-term AMS interventions have demonstrated significant reductions in MDR organism prevalence and substantial cost savings, even in resource-constrained settings [34,35,36].

5.Public and professional education campaigns

Widespread educational efforts targeting both the general population and healthcare providers are crucial to mitigate inappropriate antibiotic use. Public health campaigns should aim to increase health literacy, dispel misconceptions about antibiotic efficacy—particularly in viral infections—and reduce the demand for unnecessary prescriptions. Training modules for clinicians must emphasize responsible prescribing practices, especially in light of the misuse observed during the COVID-19 pandemic [13,14,16,25].

6.Facilitate access to non-antibiotic alternatives

National regulatory authorities and ministries of health should streamline approval processes to improve access to promising non-antibiotic therapies, including bacteriophage treatment and immunoprophylactic agents such as OM-89 and StroVac. These options represent sustainable adjuncts to traditional antibiotic therapy and are particularly valuable in cases of recurrent or MDR infections where conventional antibiotics may be ineffective or contraindicated [46,47,48,49,50,51,52].

By implementing this roadmap, Balkan countries can better align their national health policies with international AMR containment frameworks, lower healthcare expenditures, and improve patient outcomes. Tackling antimicrobial resistance in urinary tract infections is not only a microbiological challenge but also a critical policy priority—one that requires coordinated action, political commitment, and sustained investment at both national and regional levels.

### 2.6. The Ethical, Economic and Global Health Implications of Multidrug-Resistant Urinary Tract Infections (UTIs) in the Balkans

The increasing prevalence of multidrug-resistant urinary tract infections (MDR UTIs) in the Balkans constitutes a pressing clinical, public health, ethical, and economic challenge. The regional response to MDR UTIs, situated at the intersection of antimicrobial resistance (AMR), healthcare equity, and resource allocation, mirrors broader global issues, particularly for low- and middle-income countries navigating the aftermath of the COVID-19 pandemic.

In this context, several strategic considerations should be emphasized:1.Ethical Imperatives: Equity and informed decision-making

One of the most urgent ethical concerns surrounding MDR UTIs is the inequitable access to effective diagnostic tools and treatment options. In many Balkan settings, the default reliance on empirical broad-spectrum antibiotics—often initiated without microbiological confirmation—is not only clinically suboptimal but also ethically questionable [23,26]. This practice exposes patients to unnecessary toxicity, promotes further resistance, and exacerbates disparities between those who can afford rapid diagnostics or novel therapies and those who cannot. The limited use of alternative treatments such as bacteriophages and immunoprophylactic vaccines also reflects structural inequities, regulatory inertia, and a lack of clinician awareness [46,50,51]. These therapeutic options hold particular promise for high-risk populations, such as individuals with recurrent infections or prior treatment failures, yet remain inaccessible or unapproved in much of the region.

2.Economic pressure on health systems and households

MDR UTIs place a substantial financial burden on already strained public health systems. The costs associated with prolonged hospitalizations, intravenous antibiotic administration, use of second- and third-line antimicrobial agents, and repeated diagnostic procedures are significantly higher for resistant infections compared to susceptible ones [29,36]. Although antimicrobial stewardship programs and rapid diagnostics have been shown to reduce both the length of hospital stay and treatment-related expenses, their implementation requires initial investments that many Balkan healthcare systems are unable or unwilling to absorb [34,35,41].

Moreover, in several countries across Southeastern Europe, high levels of out-of-pocket spending impose an additional burden on households—especially those caring for elderly patients with recurrent infections. This often leads to delays in seeking care, resulting in more advanced disease at the time of admission and higher healthcare costs. In this way, the economic impact of MDR UTIs risks becoming increasingly unsustainable, further deepening health inequities.

3.Global Health Interdependence and Responsibility

The AMR crisis is inherently transnational in nature. Resistant pathogens do not recognize borders, and the Balkans—due to its geographic position and high population mobility—represents a potential conduit for the spread of resistance across the region. International surveillance platforms such as EARS-Net and CAESAR provide mechanisms for early detection and coordinated response; however, many Balkan countries remain underrepresented or contribute inconsistent data to these systems [21,23,24]. From a global health ethics perspective, wealthier EU member states and international public health agencies share a responsibility to support the strengthening of surveillance infrastructure, diagnostics, and stewardship capacities in the Balkans. As underscored during the COVID-19 pandemic, global health security depends on the resilience of every node in the infectious disease surveillance network. Investing in MDR UTI control in the Balkans is therefore not merely an act of solidarity—it is a pragmatic and necessary response to a shared global threat.

## 3. Conclusions

The alarming rise in multidrug-resistant urinary tract infections (MDR UTIs) across the Balkans calls for an urgent and coordinated response. The COVID-19 pandemic has both exposed and intensified pre-existing weaknesses in surveillance systems, antimicrobial stewardship, and diagnostic infrastructure throughout the region. Although a range of promising tools—such as rapid diagnostics, novel antimicrobials, and non-antibiotic alternatives—are now available, their implementation remains inconsistent and, in many cases, inaccessible. Strengthening laboratory infrastructure, enhancing regional surveillance, training healthcare professionals, and investing in stewardship programs must be prioritized. In parallel, policymakers should establish clear regulatory pathways for the integration of bacteriophage therapy and vaccine-based prevention strategies, which offer sustainable, long-term solutions. We strongly believe that a comprehensive, multi-faceted approach is imperative to effectively address the MDR UTI crisis and to build a more resilient and responsive healthcare system in the Balkan region.

## Figures and Tables

**Figure 1 antibiotics-14-00849-f001:**
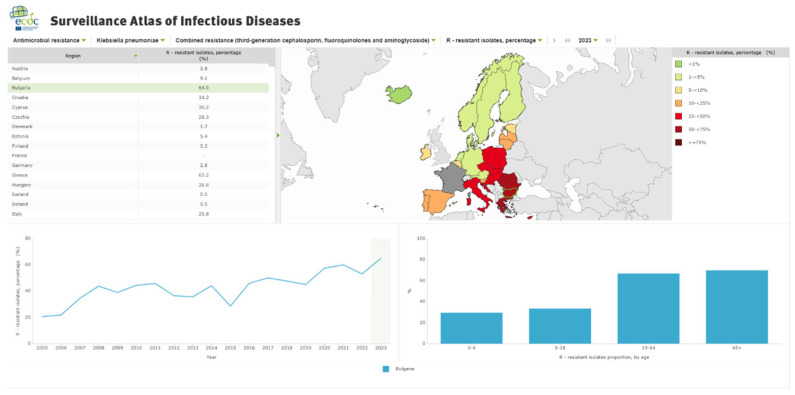
Combined resistance of Klebsiella pneumonia to third-generation cephalosporins, fluoroquinolones and aminoglycosides. (© European Centre for Disease Prevention and Control, 2025. Reproduction is authorized, provided the source is acknowledged.) (Information from: https://atlas.ecdc.europa.eu/, 5 May 2025).

## Data Availability

Data are available upon reasonable request.

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
