# Peer review of "From Pandemic to Resistance: Addressing Multidrug-Resistant Urinary Tract Infections in the Balkans"

_antibiotics, 2025, doi:10.3390/antibiotics14090849_

Round 1

Reviewer 1 Report

Comments and Suggestions for Authors

This review addresses a significant topic within a region that has been subject to limited attention. The article presents well-researched data on resistance from both the regional context and the global perspective. The recommendations provided are well-founded in antimicrobial stewardship rules and offer valuable guidance for future research or application in the practice. Overall, this is a well-written review that can serve as a resource for both specialists and the broader academic community. However, there are several comments that need to be addressed, and mistakes fixed:

Line 78: missing reference?

Lines 127-133: the study from Morocco mentioned is not cited. Also, why would you mention the resistance values of enterococci to ceftriaxone, if they are intrinsically resistant

Lines 189-191: meropenem and imipenem are carbapenem, please rephrase the sentence

Line 380: in my opinion, clinicians should be included in AMS teams, because their collaboration is necessary for these endeavours

Lines 405-406: “The ethical, economic and global health implications of multidrug-resistant urinary tract infections (UTIs) in the Balkans“– is this supposed to be a section title? (number 2.7)

Line 421: I presume there is a typing error: “bareflectages” is supposed to be “bacteriophages”

There are several problems in need of fixing with the references. Please adequately adjust website references, so they are consistent throughout and correctly done. Also, some article references have different styles, e.g. ref. 34 and 35 are different. I suggest thorough revision of the references, so the style correct.

Author Response

ANSWER

We would like to thank Reviewer 1 for the positive feedback and we highly appreciated the suggestions/comments for the improvement of our manuscript. We would like to answer them one by one.

Line 78: missing reference?

Thank you for pointing that out – it is our technical mistake and now it is corrected in the text.

Lines 127-133: the study from Morocco mentioned is not cited. Also, why would you mention the resistance values of enterococci to ceftriaxone, if they are intrinsically resistant

The paper that should be added in this section is:

El Omari, L., Sakhi, A., Miloudi, M., Elkamouni, Y., Zouhair, S., & Arsalane, L. (2023). The impact of the COVID pandemic on the uropathogenic bacterial resistance profile: Experience of the bacteriology lab of the military hospital Avicenne in Marrakech. GSC Advanced Research and Reviews, 14(2), 59–65. https://doi.org/10.30574/gscarr.2023.14.2.0045

There is technical mistake – it is not decreaseing, the resistence is increasing. Please excuse us for that!

Lines 189-191: meropenem and imipenem are carbapenem, please rephrase the sentence

It is corrected in the text! Thank you for pointing that out!

Line 380: in my opinion, clinicians should be included in AMS teams, because their collaboration is necessary for these endeavours

This is absolutely true and we have had the same discussion and idea, but when we were preparing the paper we have missed to added that essential component of the AMS team. Now it is corrected in the text, thank you!

Lines 405-406: “The ethical, economic and global health implications of multidrug-resistant urinary tract infections (UTIs) in the Balkans“– is this supposed to be a section title? (number 2.7)

Yes, it is 2.7 section title. Please excuse us for this technical mistake!

Line 421: I presume there is a typing error: “bareflectages” is supposed to be “bacteriophages”

Yes, you are absolutely right! Please excuse us for the mistake!

There are several problems in need of fixing with the references. Please adequately adjust website references, so they are consistent throughout and correctly done. Also, some article references have different styles, e.g. ref. 34 and 35 are different. I suggest thorough revision of the references, so the style correct.

Thank you for point it out! Now the reference are corrected and we have added the paper that was missed out and references are in total 54.

Reviewer 2 Report

Comments and Suggestions for Authors

Overall, the manuscript is written fluently and is easy to read. The text contains many repetitions of general phrases that could be further reworded.

Detailed comments:  

This phrase “The rise of multidrug-resistant (MDR) urinary tract infections (UTIs) presents a serious public health challenge.. "is a mental shortcut that is quite common in English scientific literature, but strictly speaking, biologically, it's not the infections themselves that are multidrug-resistant, but the bacteria that cause them. It would be more correct to write:"The rise of urinary tract infections caused by multidrug-resistant (MDR) bacteria presents a serious public health challenge…"at least at the beginning of the text

Line 44  “morbidity rates (ref).” – no ref.

Line 78 resistance (AMR) (ref). -  no ref.

Line 52-54 bacterial species names - in italics ; this also applies to subsequent names that appear

I suggest reorganizing the beginning of the article, because the information included as 2. Results 41/ 2.1 Epidemiology of MDR UTIs is general and the cited literature does not come from the Balkan countries, so it is suitable for the Introduction and not for the Results

Author Response

REVIEWER 2

Overall, the manuscript is written fluently and is easy to read. The text contains many repetitions of general phrases that could be further reworded.

Detailed comments: 

This phrase “The rise of multidrug-resistant (MDR) urinary tract infections (UTIs) presents a serious public health challenge.. "is a mental shortcut that is quite common in English scientific literature, but strictly speaking, biologically, it's not the infections themselves that are multidrug-resistant, but the bacteria that cause them. It would be more correct to write:"The rise of urinary tract infections caused by multidrug-resistant (MDR) bacteria presents a serious public health challenge…"at least at the beginning of the text.

Line 44  “morbidity rates (ref).” – no ref.

Line 78 resistance (AMR) (ref). -  no ref.

Line 52-54 bacterial species names - in italics ; this also applies to subsequent names that appear

I suggest reorganizing the beginning of the article, because the information included as 2. Results 41/ 2.1 Epidemiology of MDR UTIs is general and the cited literature does not come from the Balkan countries, so it is suitable for the Introduction and not for the Results

ANSWER

We would like to thank Reviewer 2 for the positive feedback and we highly appreciated the suggestions/comments for the improvement of our manuscript. The intention is to provide a comprehensive response to each of these points.

This phrase “The rise of multidrug-resistant (MDR) urinary tract infections (UTIs) presents a serious public health challenge.. "is a mental shortcut that is quite common in English scientific literature, but strictly speaking, biologically, it's not the infections themselves that are multidrug-resistant, but the bacteria that cause them. It would be more correct to write:"The rise of urinary tract infections caused by multidrug-resistant (MDR) bacteria presents a serious public health challenge…"at least at the beginning of the text.

We would like to thank the Reviewer for the suggestion. It is corrected in the text and we believe that the beginning is improved in this way.

Line 44  “morbidity rates (ref).” – no ref.

Please excuse us for the technical mistake. It is now improved and corrected.

Line 78 resistance (AMR) (ref). -  no ref.

It is technical mistake by our team and was missed when proofreading the text. Now it is corrected and the references were added.

Line 52-54 bacterial species names - in italics ; this also applies to subsequent names that appear

Thank you for pointing that mistake – now all the bacteria in the text are in italics.

I suggest reorganizing the beginning of the article, because the information included as 2. Results 41/ 2.1 Epidemiology of MDR UTIs is general and the cited literature does not come from the Balkan countries, so it is suitable for the Introduction and not for the Results

Thank you for the suggestion, the section is moved from Results to Introduction. We totally agree with the
